# Single-cell transcriptome of the mouse retinal pigment epithelium in response to a low-dose of doxorubicin

Hyungwoo Lee[1,2,9], Ho-Yeon Lee[3,4,9], Jae-Byoung Chae[1], Chul-Woo Park[1], Chaekyu Kim[5], Ja-Hyoung Ryu [6], Jiwon Jang [7,8], Namshin Kim [3,4✉] & Hyewon Chung [1,2✉]

Cellular senescence of the retinal pigment epithelium (RPE) is thought to play an important role in vision-threatening retinal degenerative diseases, such as age-related macular degeneration (AMD). However, the single-cell RNA profiles of control RPE tissue and RPE tissue exhibiting cellular senescence are not well known. We have analyzed the single-cell transcriptomes of control mice and mice with low-dose doxorubicin (Dox)-induced RPE senescence (Dox-RPE). Our results have identified 4 main subpopulations in the control RPE that exhibit heterogeneous biological activities and play roles in ATP synthesis, cell mobility/ differentiation, mRNA processing, and catalytic activity. In Dox-RPE mice, cellular senescence mainly occurs in the specific cluster, which has been characterized by catalytic activity in the control RPE. Furthermore, in the Dox-RPE mice, 6 genes that have not previously been associated with senescence also show altered expression in 4 clusters. Our results might serve as a useful reference for the study of control and senescent RPE.

[1] Department of Ophthalmology, Konkuk University School of Medicine, Seoul, Republic of Korea. [2] Department of Ophthalmology, Konkuk University Medical Center, Seoul, Republic of Korea. [3] Genome Editing Research Center, Korea Research Institute of Bioscience and Biotechnology, Daejeon, Republic of Korea. [4] Department of Bioinformatics, KRIBB School of Bioscience, University of Science and Technology (UST), Daejeon, Republic of Korea. [5] Fusion Biotechnology, Inc., Ulsan 44919, Republic of Korea. [6] Department of Chemistry, Ulsan National Institute of Science and Technology, Ulsan, Republic of Korea. [7] Department of Life Sciences, Pohang University of Science and Technology (POSTECH), Pohang, Republic of Korea. [8] Institute of Convergence Science, Yonsei University, Seoul, Republic of Korea. [9] These authors contributed equally: Hyungwoo Lee, Ho-Yeon Lee. ✉email: n@rna.kr; hchung@kuh.ac.kr

The retinal pigment epithelium (RPE) is a monolayer of cells located between the retinal photoreceptors and the choriocapillaris, and it plays a vital role in maintaining both the photoreceptor and the choroid in healthy conditions[1,2]. Its tight junctions protect the neuroretina, and its pigment components reduce photooxidative stress. Additionally, the RPE delivers nutrients to photoreceptors and processes waste products from the photoreceptor outer segments. The RPE plays crucial roles in the recycling of retinoids to maintain the visual cycle[3]. Additionally, the vascular endothelial growth factor secreted by the RPE maintains the function of the choriocapillaris by stimulating the formation of fenestrations[4].

Age-related macular degeneration (AMD) is the leading cause of vision loss in developed countries and is accompanied by progressive degeneration of the RPE, retina, and choriocapillaris[5]. Dysfunction of the RPE resulting from oxidative stress, impairment of nutrient delivery, energy metabolism and complement dysregulation contributes to the pathogenesis of AMD. Additionally, cellular senescence of the RPE is also considered to be an important pathologic change closely related to the development of AMD[6,7].

Cellular senescence refers to an irreversible state with arrested cell proliferation[8]. Additionally, this state involves the expression of senescence-associated β-galactosidase (SA-β-Gal), chromatin condensation, and increased expression of matrix reconstruction proteins, chemokines and inflammatory factors known as the senescence-associated secretory phenotype (SASP). Previous studies suggested that cellular senescence is involved or plays a causative role in aging and age-related diseases, including AMD[9,10]. SASP represents a highly plastic phenotype and can be a key player in senescence-related AMD pathophysiology because SASP-derived factors initiate or potentiate low-grade inflammatory processes, which are particularly relevant to aging and aging-associated diseases[11,12]. Additionally, the occurrence of senescence is consistent with the signs of AMD, including increased oxidative stress, DNA damage, decreased proteolysis and even an increase in vascular endothelial growth factor secretion in the RPE[13]. Therefore, deciphering the expression profiles of the RPE as well as the senescent RPE in more detail is important to understand the pathophysiology of AMD and might provide a new therapeutic target to treat AMD.

The RPE is known to exhibit heterogeneity, and it displays regional differences in enzymatic activities, melanin and lipofuscin granules, and morphology[14–17]. Furthermore, in AMD, pathologic changes, including drusen deposition, atrophy and choroidal neovascularization, are unevenly distributed, even in the macular area[5]. To date, gene expression studies on the RPE have collected mRNA from pooled RPE samples[18,19]. However, these studies could not guarantee the contamination of cells other than RPEs. Recent advances in single-cell RNA sequencing (scRNA-seq) have enabled the identification of cell types based on the specific cell type markers[20]. Recently, Voigt et al. conducted scRNA-seq analysis of pooled RPE and choroid tissues from human donors and identified the expression profiles of each type of cells[21]. Because the cells from choroidal tissues overwhelmed the RPEs, approximately 300 RPE cells were analyzed in this study. To enrich the number of RPEs and obtain more reliable information, enzymatic dissociation of the RPE and enrichment was applied, thereby reducing the other cell types in the prepared cells[22]. Based on these pooled RPEs, the scRNA-seq technique might reveal a more reliable RNA profile and the innate heterogeneity of RNA expression in the RPE.

Meanwhile, little is known about the specific changes that occur in subpopulations of the RPE when senescence occurs. Because few retinal pigment epithelial cells can be obtained from human donors, we speculated that scRNA-seq analysis of the larger RPE population might be appropriate to address the heterogeneous composition of the RPE and determine what subpopulation might be changed when cellular senescence is induced. Therefore, a mouse model was speculated to be a good candidate for scRNA-seq studies on the RPE.

In addition, an appropriate model of mouse RPE senescence is urgently needed. Doxorubicin (Dox) is a commonly used drug for chemotherapy that not only induces the formation of superoxide anions but also causes DNA damage and cell apoptosis[23]. A low dose of Dox is also known to promote senescence[24,25]. We recently established and validated an in vivo mouse model of RPE senescence through subretinal injection of Dox[26].

Here, we isolated cells from the RPE layers of control mice and subretinal Dox-injected mice. Using scRNA-seq, the pure RPE population was further enriched by excluding contaminant cells, and the RPE populations from control (control RPE) and Dox-injected (Dox-RPE) mice were analyzed using scRNA-seq. Characteristic expression profiles of heterogeneous subpopulations from the control RPE were analyzed and compared with those of the Dox-RPE.

## Results

**scRNA-seq analysis of prepared tissue samples.** The main purpose of the first analysis was to isolate a population of RPE cells. First, we obtained an overview of the diverse populations from all collected samples. Analysis of the 35,598 cells that passed our quality controls yielded a uniform manifold approximation and projection (UMAP) plot with 12 clusters with distinct expression profiles, even when the cells from the RPE layer were targeted for isolation (Fig. 1a, b). Then, the major cell type of each cluster was identified by comparing the expression of known specific markers of cell types expressing representative genes of each cluster (Fig. 1c and Supplementary Data 1)[21]. Among all cells, 3228 RPE cells were identified according to the expression of *Rpe65* (control RPE: 1531, Dox-RPE: 1697, Table 1), and clusters 3 and 5 were mainly composed of RPE cells (Fig. 1a, c). The other cells mainly corresponded to known cell types in choroidal tissues[21]. Fibroblasts were most abundant and were represented as clusters 1, 4 and 11. Other cells known to exist in the choroid, including melanocytes, macrophages, NK cells and choroidal vessel-associated cells—smooth muscle and endothelial cells—were mainly distributed in clusters 2, 6, 8, 7, and 9, respectively (Fig. 1a, c)[21]. The scRNA-seq results showed that the cells collected from the RPE layers included other cell types from choroidal tissues, and RPE cells could be isolated based on significantly higher expression of the *Rpe65* gene.

**Functional signatures and heterogeneity of the control RPE population.** To reveal the heterogeneity of cells within the RPE, we performed another round of clustering on the cells with high Rpe65 expression. Control RPE cells were further subdivided into 5 clusters in the UMAP plot (Fig. 2a, Supplementary Fig. 1), which shows the differently expressed genes (Fig. 2b and Supplementary Table 1).

To investigate whether specific functions could be assigned to each cluster, Gene Ontology (GO) analyses were performed using the top 100 representative markers of each cluster (Fig. 2c, Supplementary Data 2). The results showed that metabolic processes, including ATP synthesis (cluster 1), cell mobility and differentiation (cluster 2), mRNA processing and splicing (cluster 3), and catalytic processes (cluster 4), were enriched for four of the clusters (Fig. 2c). Cluster 5 consisted of 19 small cells and was associated with the muscle system and cGMP-mediated signaling. (Fig. 2c). The enrichment of these distinct biological processes revealed the possible functional heterogeneity of control RPE cells in vivo.

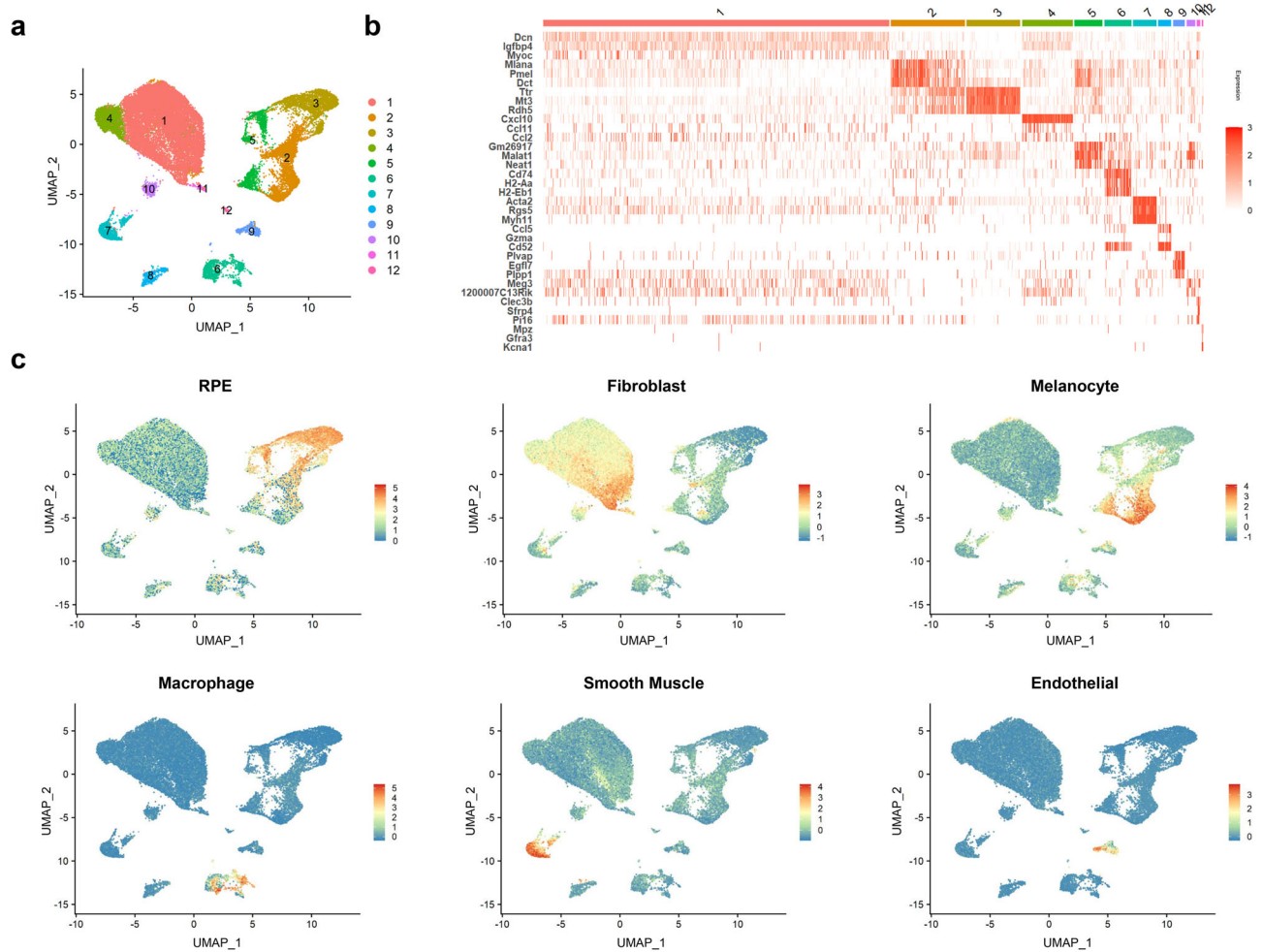

**Fig. 1 scRNA-seq analysis of whole isolated cells yielded 12 distinct cell populations. a** UMAP plot depicting the single-cell transcriptomes of whole cells from prepared RPE tissues from control and Dox-treated mice. Each dot represents a single cell ($n = 35{,}598$). Colorization was performed according to the unsupervised clustering performed by Seurat. **b** Heatmap showing the three most DEGs of each cell cluster, as provided by Seurat. Each column represents a single cell, and each row represents an individual gene. Three marker genes per cluster are shown on the left, sorted by $p$ value. Red indicates maximum gene expression, and white indicates no expression in scaled log-normalized UMI counts. **c** Average expression of well-established cell type markers was projected on the UMAP plot to identify all cell populations (see the "Methods" for details). Red indicates maximum gene expression, while blue indicates low or no expression of a particular set of genes in log-normalized UMI counts.

**Table 1 Number of cells in each cluster of control retinal pigment epithelium cells and low-dose doxorubicin-treated cells.**

| Group | Cluster 1 | Cluster 2 | Cluster 3 | Cluster 4 | Cluster 5 | Total |
|---|---|---|---|---|---|---|
| Control | 871 | 331 | 213 | 98 | 18 | 1531 |
| Dox | 954 | 216 | 277 | 244 | 6 | 1697 |

Control = control mouse retinal pigment epithelium (RPE) cells; Dox = mouse RPE cells treated with low-dose doxorubicin.

**Low-dose Dox regulates apoptosis and increases senescence in the RPE**. Next, we investigated the effects of low-dose Dox based on the single-cell transcriptome of Dox-RPE cells. In the UMAP plot, Dox-RPE cells also showed heterogeneity, as they could be divided into 5 clusters, and the distribution of each cluster in the UMAP plot (Fig. 3a) and the most representative expressed genes of each cluster were similar to those for control RPE cells (Fig. 3b and Supplementary Table 2). These similarities in heterogeneity and the most highly expressed genes between control and Dox-RPE cells might reflect the limited effect Dox in vivo, as subretinally injected Dox usually reaches one fourth of the total RPE area.

Interestingly, GO biological process analyses of the top 100 representative markers of each cluster in Dox-RPE cells suggested

a considerable change in functional annotation, especially in cluster 4 (Fig. 3c, Supplementary Data 3). Cluster 4 in the control RPE was originally characterized by catalytic process and associated functions (Fig. 2c). On the other hand, Dox-induced counterparts in cluster 4 showed enrichment of GO processes associated with apoptosis (Fig. 3c). In the other clusters (1, 2, 3, and 5), similar GO processes were found in both control- and Dox-induced RPE cells (Figs. 2c and 3c).

Using Ingenuity Pathway Analysis (IPA), we also examined the canonical pathway for the each cluster of control RPE and Dox-RPE. Our analysis revealed that the 5 subclusters showed distinct pathways, as well as general overlap of pathways between each cluster of control and Dox-RPE in all clusters (Supplementary

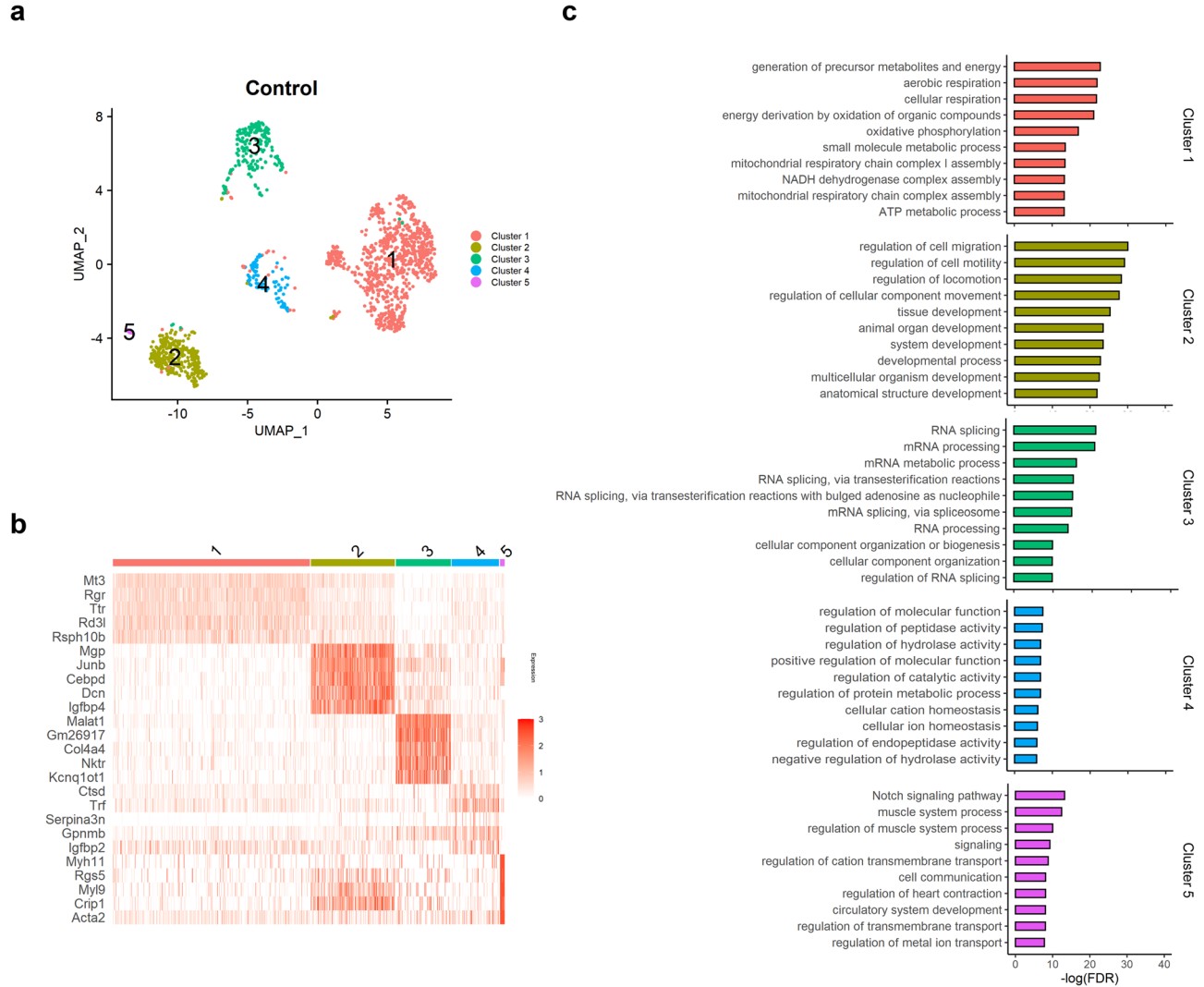

**Fig. 2 RPE subpopulations in control mice display specific functional transcriptomic signatures. a** UMAP plot displaying RPE cells from control mice ($n = 5$). Each dot represents a single cell ($n = 1531$). Colorization was performed according to the original unsupervised clustering performed by Seurat. **b** Heatmap showing the three most DEGs of each cell cluster, as provided by Seurat. Each column represents a single cell, and each row represents an individual gene. Five marker genes per cluster are shown on the left, sorted by $p$ value. Red indicates maximum gene expression, and white indicates no expression in scaled log-normalized UMI counts. **c** Top 10 enriched GO terms in each control RPE subpopulation, sorted by $p$ value. Control = control mouse RPE cells.

Fig. 2). An apoptosis associated pathway was only found in cluster 4 of Dox-RPE, demonstrating again that cluster 4 of Dox-RPE represents senescence-specific gene expression.

To further characterize the apoptosis-associated change in the Dox-RPE in more detail, 187 differentially expressed genes (DEGs) were identified, which showed statistically significant differences between control- and Dox-RPE cells in any of the 5 clusters (Supplementary Data 4, Supplementary Figs. 3 and 4). Genes associated with the negative regulation of apoptosis were elevated in Dox-RPE cells: *Ckmt1, Ccng1* and *Gas6* (Fig. 4). Among proapoptotic genes, the *Bad, Bak1*, and caspase levels were not significantly changed, while the *Bax* and *Aen* levels were elevated in cluster 4 (Fig. 4a). These data show that the apoptotic process is generally negatively regulated in our in vivo senescence model[27,28].

Senescence-associated changes were also observed in Dox-RPE cells, mainly in cluster 4. The expression of *Cdkn1a*, a representative marker of cellular senescence, was significantly elevated in cluster 4 of Dox-RPE cells (Fig. 4b). The levels of *B2m* and *A2m* encoding alpha- and beta-2 macroglobulin, respectively,

which are also associated with cellular senescence, were also elevated in the same cluster (Fig. 4b)[29,30]. The expression of *Ntrk2*, which encodes neurotrophic receptor tyrosine kinase 2, was also elevated in cluster 4 and elevated in the brains of senescence-accelerated mice[31]. Furthermore, the expression of *Timp1*, which is known to maintain a senescent state in prostate tumors[32], was increased in clusters 4 and 5 (Fig. 4b). Overall, these results reflect the apoptosis-resistant phenotype of cellular senescence in vivo when the RPE was treated with Dox.

Interestingly, among DEGs, *Ednrb, Pmel, S100b, Selenop*, and *Spp1* showed decreased levels in all clusters except for cluster 5 after Dox treatment, while *Serpina3n* levels increased (Fig. 4c). These changes spanning all clusters are intriguing because the expression of the genes related to senescence or the regulation of apoptosis mentioned above were changed mainly in cluster 4. Furthermore, the association of these 6 genes with Dox treatment or senescence is not well known, especially in vivo in the RPE.

To further investigate the biological processes not directly related to apoptosis or senescence, we performed GO analyses of the DEGs (up- and downregulated) (Fig. 5a, b). The results

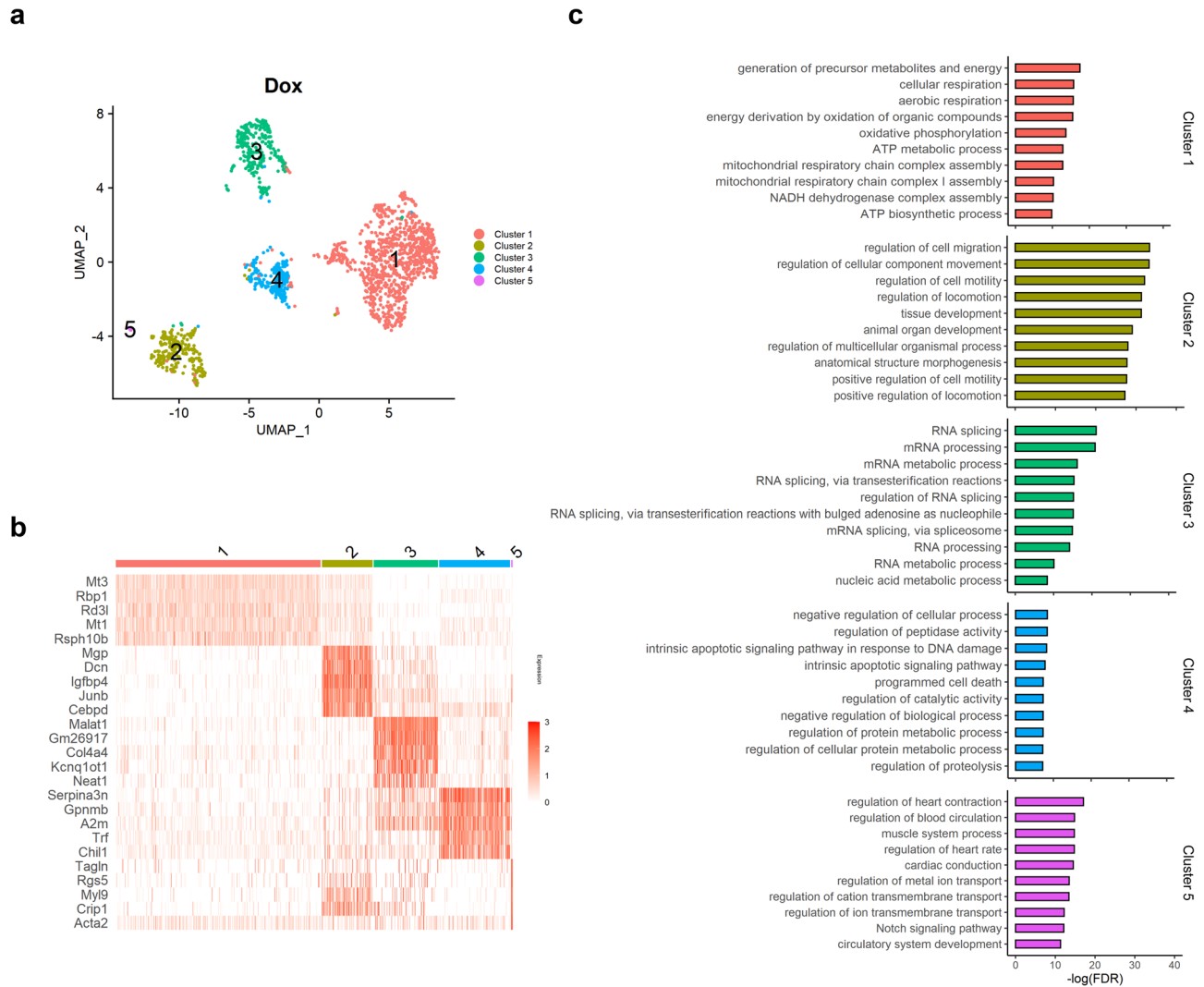

**Fig. 3 Dox treatment regulates apoptosis and increases senescence in the RPE. a** UMAP plot displaying RPE cells from Dox-treated mice ($n = 5$). Each dot represents a single cell ($n = 1697$). Colorization was performed according to the original unsupervised clustering performed by Seurat. **b** Heatmap showing the three most DEGs of each cell cluster, as determined by Seurat. Each column represents a single cell, and each row represents an individual gene. Five marker genes per cluster are shown on the left, sorted by $p$ value. Red indicates maximum gene expression, and white indicates no expression in scaled log-normalized UMI counts. **c** Top 10 enriched GO terms in each Dox-treated RPE subpopulation sorted by $p$ value. Dox = mouse RPE cells treated with low-dose Dox.

suggested that the responses to interferon-beta, viruses, organic substances and cytokines were significantly enhanced, while translation and biosynthetic processes were downregulated after Dox treatment. These results might reflect increased inflammation and cell cycle arrest.

**Validation of RPE subpopulations using RNA fluorescence in situ hybridization**. To further characterize and validate the RPE subpopulations from our initial analysis, we identified the most representative markers for each subpopulation according to their expression in the specific cell clusters. We then performed RNA fluorescence in situ hybridization (FISH) on the flattened mouse RPE tissue (RPE flatmount), which can show the 2-dimensional representation of RPE cells in control and Dox-treated mice. *Rdh5, Serping1*, and *Malat1* were selected markers for RPE clusters 1, 2, and 3, respectively (Supplementary Table 1). These markers were the common cluster markers of control and Dox-RPE. For cluster 4, we selected Dox-RPE specific markers representing apoptosis (*Bax*) and senescence (*Cdkn1a, A2m*) to additionally validate the Dox-RPE specific expression. Because of

the paucity of RPE in cluster 5 in both control and Dox-RPE cells, we validated the expression and spatial expression of clusters 1–4 only.

In result, *Rdh5, Serping1, and Malat1* showed differences in the spatial distribution among one another (Fig. 6, Supplementary Fig. 5). This spatial distribution was observed in the same pattern at both phosphate-buffered saline PBS or Dox injected areas and in non-injected areas from the two groups (Control RPE and Dox-RPE). Three markers of cluster 4 (*Bax1, Cdkn1a, A2m*) showed similar spatial distributions in the Dox injected area in Dox-RPE, as expected. Conversely, they showed a significantly different spatial distributions from the other markers of clusters 1, 2, and 3 (Fig. 6, Supplementary Fig. 5).

Together, these results provide validation of the results from scRNA-seq and establish markers for the detection of specific RPE subpopulations.

**Discussion**
ScRNA-seq has emerged as a powerful tool to study gene expression in heterogeneous tissues such as the choroid. As

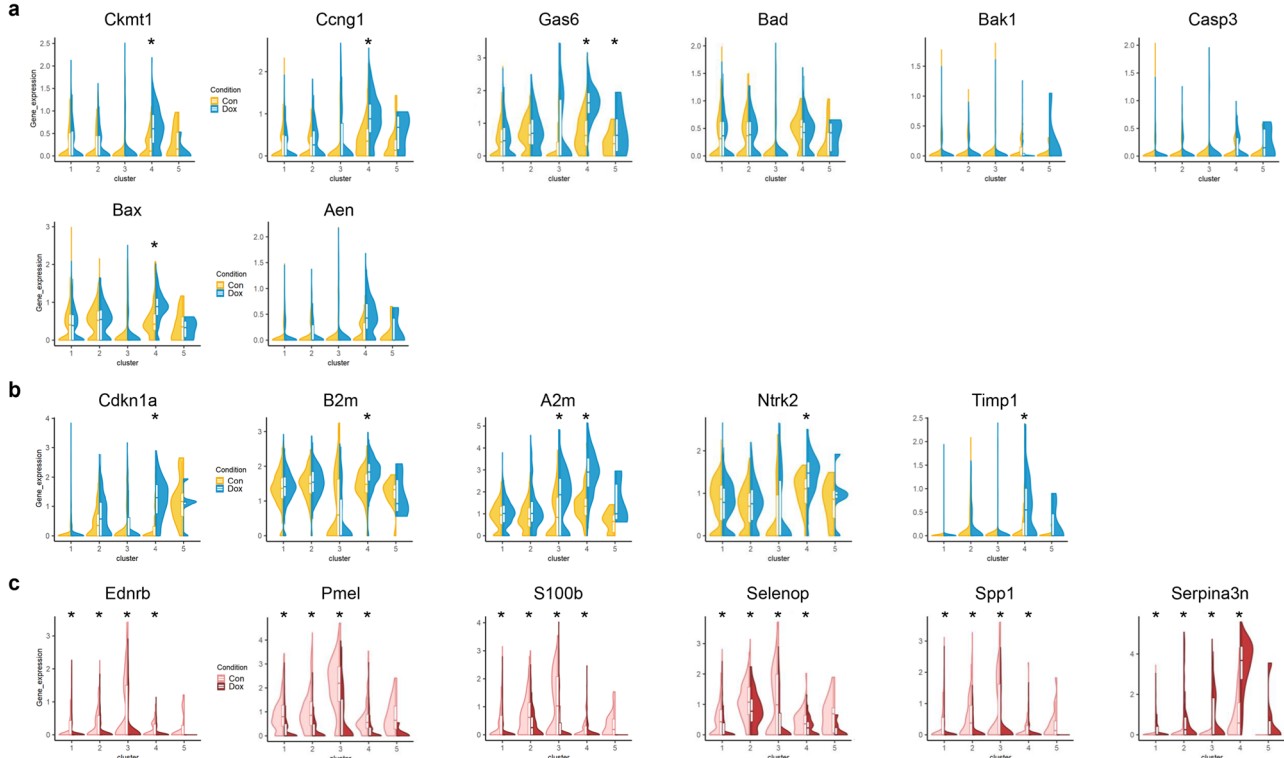

**Fig. 4 Dox treatment-induced DEGs are associated with apoptosis/senescence in a specific cluster while 6 genes are changed in all clusters. a** Violin plots of genes associated with the regulation of apoptosis. **b** Violin plots of the genes associated with cellular senescence. **c** Violin plots of the genes that showed significant changes in expression in all 5 clusters. The X-axes depict the cell cluster number, and the Y-axes represent the expression of each gene. The statistical significance of differences in expression between control and Dox-treated RPE cells is indicated as an asterisk on the corresponding cluster based on the p values of the corresponding Wilcoxon rank sum tests.

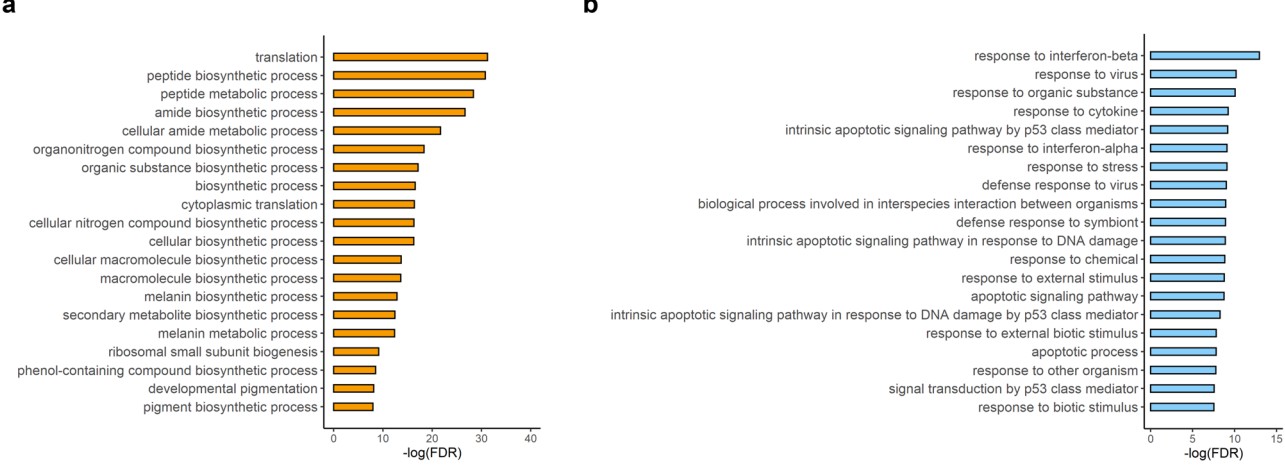

**Fig. 5 Other Dox-related biological processes in the RPE based on upregulated and downregulated DEGs. a** Bar graph showing the top 20 enriched GO terms for the upregulated DEGs in all clusters. **b** Bar graph showing the top 20 enriched GO terms for the downregulated DEGs in all clusters.

opposed to bulk RNA sequencing, where transcripts from different cell types are aggregated and analyzed together, this technique allows for precise mapping of gene expression patterns to individual cell types[33,34]. Previous studies established the single-cell transcriptomic atlas of the RPE and the adjacent choroid in nonhuman primates and human donor eyes[21]. These studies revealed the global landscape of expression profiles from multiple tissues, but the number of RPE cells was as low as hundreds, since the RPE constitutes a monolayer between the retina and the choroid, which have much greater volume and cell numbers when they are obtained together. To overcome this limitation, we collected 3228 mouse RPE cells in total by enriching RPE cells as much as possible by meticulously dissecting the adjacent retina and choroidal tissue as well as implementing quality control processes to confirm the existence of a proper RPE population. Using prepared whole cells, we found transcripts specific for cell populations including RPE cells and endothelial cells, melanocytes, fibroblasts, and leukocytes from the choroid. While other cell types were found, we analyzed a larger RPE population than previous studies[21,35].

From the UMAP plot of control RPE cells, we found heterogeneity of RPE cells in 5 subgroups. While the heterogeneity of

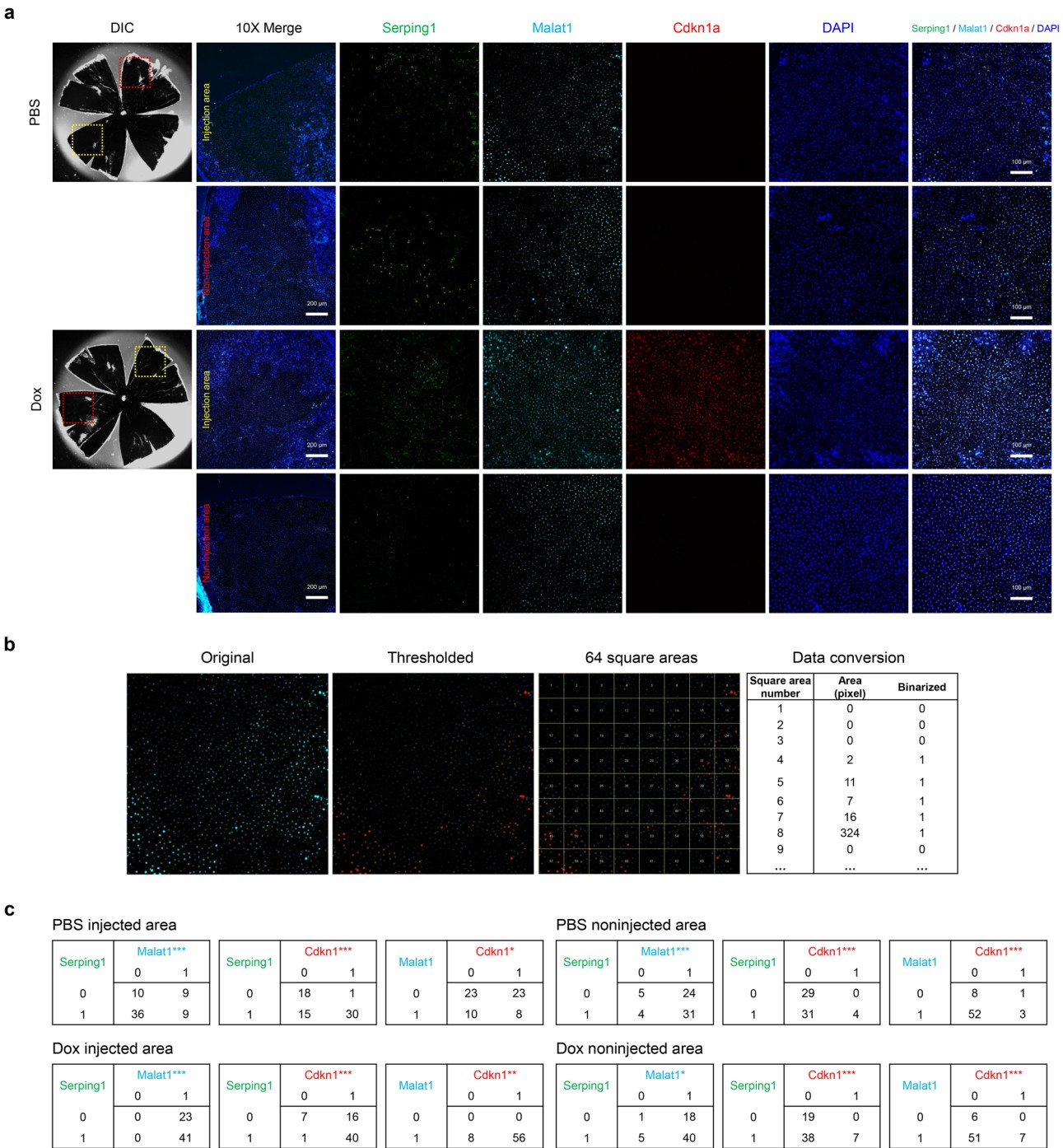

**Fig. 6 RNA fluorescence in situ hybridization of RPE subpopulations in control RPE and Dox-RPE. a** Images showing mRNA expression of *Serping1* (green), *Malat1* (cyan) and *Cdkn1a* (red), selected markers for RPE subpopulations 2, 3, and 4, respectively. Nuclei were counterstained with DAPI. Differential interference contrast (DIC) images represent the whole shape of the RPE flatmount and the inspected location in the injected area (yellow box) and noninjected area (red box). **b** Image processing to represent spatial distributions as series of numbers. The original image was binarized by an arbitrarily determined threshold to represent the spatial distribution of a marker, and the whole image was divided into 64 square areas. Then, the information about expression in each square area was marked as 1 or 0 according to the presence or absence of a signal inside. **c** 2 × 2 contingency tables between 2 markers in four environments of the same RPE flatmount in PBS-injected/ noninjected areas, Dox injected/noninjected areas. Statistical analyses were performed using McNemar's test (*$p < 0.05$, **$p < 0.01$, ***$p < 0.001$). Scale bar: 200 μm for the 10× merge image (first column) and 100 μm for ×20 magnification (next columns on the right side) in **a**. PBS = RPE flatmount from the PBS injected mouse; Dox = RPE flatmount from the Dox-injected mouse.

RNA expression in RPE cells has not been studied well, there have been many studies on topographic heterogeneity comparing the features of the RPE to uncover regional heterogeneity (posterior vs. peripheral location) and cell-cell heterogeneity[14]. For regional heterogeneity, morphometric analyses demonstrated topographical differences in RPE size, shape, density, and granule content. Additionally, posterior RPE cells had a tendency to have a lower growth potential in vitro[36], lower activities of some lysosomal enzymes, and higher activity of cathepsin D. Regarding cell-cell heterogeneity, individual cells or patches showed differences in the number of melanosomes and lipofuscin granules[16]. Additionally, Sonoi et al. reported the phenotypic heterogeneity of cultured human RPE cells according to their nuclear size and local density[17]. In bovine RPE cells, cell-to-cell variability in enzymatic activities was reported[15]. In our results, ATP synthesis (cluster 1), cell mobility and differentiation (cluster 2), mRNA processing and splicing (cluster 3), and catalytic processes (cluster 4), were enriched in four of the clusters (Fig. 2c). Cluster 5 was associated with the muscle contraction and cation transport. These results are not directly consistent with the previous results of regional or cell–cell heterogeneity in terms of morphology, abundance of granules, or enzymatic activity. Rather than these aspects, our results are the first to suggest the existence of pronounced heterogeneity in vital activities, including energy generation, motility, RNA processing, and catalytic process in mouse RPE cells. As in the human retina, the central region of the mouse retina possesses a higher photoreceptor cell density and a thinner Bruch's membrane than the periphery, which underlies the higher phagocytic load of the mouse RPE[37]. This difference might have resulted in the pronounced cluster of ATP synthesis (clusters 1, Fig. 2c). Because we did not divide the posterior and peripheral areas of the RPE, further study with topographic separation might reveal the ATP synthesis activity according to the lesion. The function of cluster 2 was represented by cell motility. Quiescent RPE cells rests on a collagen type IV- and laminin-containing basement membrane on the inner aspect of Bruch's membrane, and they do not normally divide or migrate away from this layer[3]. Mechanical injury caused by subretinal PBS injection might induce increased RPE cell motility. Chorioretinal injuries might expose RPE cells to a variety of cytokines and inflammatory cells, resulting in RPE cell activation and separation from the monolayer[38,39]. The reason why mRNA processing is prominent in a specific cluster (cluster 3) is not clear. The GO process term 'mRNA processing' is defined as any process involved in the conversion of a primary mRNA transcript into mature mRNA. Therefore, cluster 3 is thought to represent active mRNA maturation, mRNA processing and subsequent protein synthesis. Additionally, cluster 3 showed increased melanin biosynthetic processes (Supplementary Data 2). Therefore, the previously reported granule heterogeneity of the RPE can be explained by our observation of heterogeneity in RNA expression[16]. Cluster 5 in both PBS and Dox-RPE was associated with muscle contraction processes, while the number of cells were quite lower than other clusters. The muscle contraction and upregulated smooth muscle associated genes—Myl9 and Mylk—in cluster 5 might be related to another major cellular process, such as epithelial-to-mesenchymal transition (EMT) of the RPE. Active modulation of the cytoskeleton, including increased contractility, was associated with EMT[40]. In addition, Acta2 and Myl9, two marker genes of cluster 5, have also been reported in cells other than muscle cells, such as the subpopulation of the human pluripotent stem cells[41].

In Dox-RPE cells, negative regulation of the apoptosis process was prominent (Fig. 4a), and senescence-associated gene expression was increased (Fig. 4b), especially in cluster 4. The expression of Cdkn1a and other senescence-associated genes— A2m, B2m, and Ntrk2—was increased in the same cluster. These

results reflect the apoptosis-resistant phenotype of cellular senescence in the Dox-RPE[27,28]. Moreover, the validity of the in vivo mouse model of subretinal Dox injection was confirmed by the RNA profiles at the single-cell level, which was previously confirmed by the change in p53/p21 expression and the elevated SA-β-Gal levels[26]. In contrast to cluster 4, the GO processes of other clusters (clusters 1, 2, 3, 5) were similar to those of the control RPE. Furthermore, the major processes of cluster 4 in the control RPE and Dox-RPE were catalytic processes, including peptidase and hydrolase activities. Therefore, we can speculate that the senescent RPE might be closely associated with the specific cell population that is primarily associated with proteolytic processes. Imbalance of proteostasis in aging is closely associated with protein aggregation, accumulation of misfolded proteins and, in the end, cellular dysfunction[42]. Our result is intriguing in that the low-dose doxorubicin treatment did not affect all RPE cells but was mainly associated with the specific RPE subpopulation characterized by the catalytic activity. This specific RPE subpopulation could be more susceptible to the Dox treatment because their main activity is focused on the regulation of catalysis, including proteolysis, leading to the senescence phenotype.

Overall, the subclusters found in control mouse RPE cells represent functional cell-cell heterogeneity. In addition, we also found that RPE subpopulations had distinct spatial localizations in the RPE flatmount through the RNA fluorescence in situ hybridization (FISH) experiments in control mice, validating the results from scRNA-seq. Therefore, the functional and spatial heterogeneity of RPE cells was demonstrated even in a quite small microscopically defined area, although RPE cells in this area looked similar to the dark pigmented and hexagonal shape. In subretinally Dox-injected mice, the same marker genes of clusters 1, 2, and 3 in both Dox injected and non-injected areas also showed similar heterogeneity to those in control mice, consistent with scRNA-seq results, which showed similar biological process between control and Dox-injected RPE in these clusters. The marker genes related to apoptosis and senescence specific to cluster 4 showed an increased signal in only in the Dox-injected area, validating the increase in cellular senescence processes localized to the area that was affected by the subretinal injection of low-dose Dox. The SA-β-Gal positive area on gross microscopic examination was definitely composed of cells expressing cellular senescence traits[26]. At the same time, based on the present RNA FISH results, researchers should aware that the RPE cells undergoing different biological processes (clusters 1, 2, and 3) are heterogeneously distributed in Dox-injected areas.

Based on the 187 DEGs whose expression levels were significantly different in any cluster, we further characterized the other GO processes that were not designated apoptosis or senescence (Fig. 5a, b). In the Dox-RPE, responses to interferon-beta, viruses, organic substances and cytokines were significantly increased. Chemotherapeutics such as Dox and cisplatin have been shown to increase the secretion of cytokines such as IL-6 and IL-8, as well as other molecules from tumor cells, and this phenomenon is closely related to the induction of type I interferons—antiviral cytokines including interferon alpha and beta. Therefore, these processes might reflect the inflammatory status after Dox treatment. Translation and biosynthetic processes were decreased after Dox treatment. These results reflect the arrested cell cycle in the senescent state.

Interestingly, among these DEGs, Ednrb, S100b, Selenop, Spp1, and Pmel showed decreased expression in all 5 clusters in Dox-RPE cells, while Serpina3n expression was increased in all 5 clusters (Fig. 4c). These 6 genes were intriguing in that their expression changed in all clusters, while the genes related to senescence or the regulation of apoptosis were changed mainly in

cluster 4. In the RPE, the roles of EDNRB (endothelin receptor type B), S100B (S100 calcium binding protein B), SELENOP (selenoprotein P), SPP1 (secreted phosphoprotein 1), and Serpin A3N (serine protease inhibitor A3N) are not clear, and the association with Dox has not yet been elucidated. PMEL plays an essential role in the structural organization of premelanosomes[43]. The RPE is densely packed with melanosomes containing melanin pigment, which can reduce harmful backscattered light and remove free radicals that arise during phagocytosis of photoreceptor outer segments[3]. The decrease in PMEL in Dox-treated RPE cells might be associated with a previous finding related to the loss of melanin granules in old RPE cells[44]. Interestingly, decreased SPP1 expression was concordant with attenuated secreted levels in the stem cell niche upon aging[45]. Furthermore, treating aged hematopoietic stem cells with thrombin-cleaved SPP1 led to phenotypic and functional rejuvenation. Therefore, further study is needed to elucidate the role of these 6 genes in the senescence of the RPE. Therefore, these genes might be new markers of senescence in the RPE that are distinct from the classic markers in that their expression change is global rather than restricted to a specific population such as cluster 4.

This study has several limitations. Subretinal injection of Dox did not affect the entire RPE but affected approximately one-fourth of the RPE; therefore, unlike in in vitro models, in which a drug affects the entire cell, a drug effects on only a portion of cells in this model. Additionally, Dox treatment induces a relatively faster change in senescence, but AMD is a slowly progressive disease in patients. Since no animal models so far have recapitulated all of the pathologies of human AMD, the scRNA-seq data in this study might not fully reflect the real transcriptomic change in AMD patients. Therefore, further studies are needed in the chronic model, such as comparing the transcriptome of young mice with that of old aged mice. Furthermore, comparing the similarity and difference in gene expression among various known animal models in the single cell level might be also beneficial to correctly interpreting the experimental results from each animal model and to understanding the pathology at the molecular level.

In conclusion, this study enhanced our understanding of the expression profile of RPE cells at the single-cell level by enriching a large number of RPE cells. In the RPE cells from control mice, we found heterogeneity, and the cells were categorized into 5 subgroups. From the Dox-treated in vivo mouse model, we confirmed the antiapoptotic and senescent changes in the expression profiles, as expected. Furthermore, we identified the changes in genes that had not been recognized as related to senescence. The RPE-specific transcriptome in this study has afforded us a broader understanding of the role of RPE senescence in AMD and will therefore be useful for developing treatments targeting RPE senescence and testing their effectiveness.

## Methods

**Experimental animals**. Mice were maintained in accordance with the guidelines established by the Konkuk University Institutional Animal Care and Use Committee (IACUC) and housed in a controlled barrier facility within the Konkuk University Laboratory Animal Research Center. All experimental and animal care procedures were conducted according to guidelines approved by the Konkuk University IACUC (KU IACUC; approval No. KU19015). Ten C57BL/6J male mice (aged 3 months) were purchased from Charles River Laboratories Japan Inc. (Yokohama, Japan) and were allowed to acclimate to the facility for one week before the experiments. For the scRNA-seq experiments, 5 mice (10 eyes) were randomly assigned to each control or Dox subretinal injection group. All animal experiments were conducted in a blinded manner. The mice were anesthetized with a mixture of Zoletil (Carros, France) and xylazine (Leverkusen, Germany) (4:1, diluted with normal saline), and their pupils were dilated with topical Tropherine eye drops (single use, phenylephrine hydrochloride (5 mg/ml) and tropicamide (5 mg/ml), Hanmi Pharm, Seoul, Korea). An antibiotic ophthalmic ointment (Tarivid, Santen, Osaka, Japan) was applied to all eyes after the procedures.

**Subretinal injection**. Under an optical microscope (Olympus SZ51, Tokyo, Japan), a small hole was created at the limbus of both eyes with a 30-gauge sterile needle (BD Science, San Jose, USA). Through the hole, a blunt 35-gauge Hamilton microsyringe (Hamilton Company, NV, USA) was inserted slowly. One microliter of 100 ng/μl Dox or PBS was injected into the subretinal space of C57BL/6J mice.

**Eyeball isolation and single-cell dissociation of cells in the RPE layer**. Seven days after subretinal injection of Dox or PBS, the mice were anesthetized with a mixture of Zoletil (Carros, France) and xylazine (Leverkusen, Germany) (4:1, diluted with normal saline), and their eyes were immediately enucleated. The anterior eye cups were quickly dissected and placed in cold PBS. After carefully removing the retina, RPE/choroid/scleral complex tissues were dissociated immediately using the papain dissociation system (Worthington, Lakewood, NJ, USA) following the manufacturer's instructions. Briefly, RPE/choroid scleral complex tissues were incubated with a papain (20 U/ml) solution supplemented with DNase-I (2000 U/ml) in Earle's balanced salt solution (EBSS) at 37 °C in a water bath for 30 min. RPE/choroid/scleral complex tissues were triturated with a serological pipette every 5 min. The dissociated single RPE cell pellet and potential contaminants composed of other cell types was washed with PBS and collected after centrifugation at 1200 RPM for 3 min. RPE cells were stained with 0.4% trypan blue, and pigmented single cells were manually counted using a hemocytometer.

**Single-cell RNA-Seq**. Suspended cells were filtered out using a 30 μm cell strainer (Miltenyi Biotech, cat no. 130-098-458) and washed two times with cold $Ca^{2+}$- and $Mg^{2+}$-free 0.04% BSA/PBS at 300 g for 5 min at 4 °C. After the dead centrifugation, the supernatant was removed, and dead cells were removed using a Dead Cell Removal Kit (Miltenyi Biotech, cat no. 130-090-101) and MS columns (Miltenyi Biotech, cat no. 130-042-201) according to the manufacturer's protocol. Then, the collected live cells were gently resuspended in 1 ml of ice-cold 0.04% BSA/PBS and counted with a LUNA-FX7™ Automated Fluorescence Cell Counter (Logos Biosystems). According to the 10x Chromium Single Cell 5′ v2 protocol (10x Genomics, document no. CG000331), scRNA-seq libraries were prepared using the Chromium controller and Next Gem Single cell 5′ Reagent v2 kits (10x Genomics). Briefly, the cell suspensions were diluted in nuclease-free water to achieve a targeted cell count of 10,000. The cell suspension was mixed with reverse transcription master mix and loaded with Single Cell 5′ Gel Beads and Partitioning Oil into a Single Cell K Chip. RNA transcripts from single cells were uniquely barcoded and reverse-transcribed within droplets. cDNA molecules were pooled and enriched via PCR. For the 5′ gene expression library, the amplified cDNA was sequentially subjected to fragmentation, end repair, A-tailing, and ligation of the adapters. The products were amplified via PCR to create the 5′ gene expression library. The purified libraries were quantified using qPCR according to the qPCR Quantification Protocol Guide (KAPA) and checked for quality using the Agilent Technologies 4200 TapeStation (Agilent Technologies). Then, the libraries were sequenced using the HiSeq platform (Illumina), and 150 bp paired-end reads were generated. The sequencing depth of the 5′ gene expression library was approximately 20,000 read pairs per cell. We applied Cell Ranger (v6.0.0) for standard analysis of raw read data. FASTQ files were processed using the Cell Ranger pipeline to obtain a gene-by-cell expression matrix based on the mm10 reference genome. Cell Ranger was used for sample demultiplexing, barcode processing, single-cell 5′ gene counting and data analysis.

**Data processing**. Raw sequencing data were processed through CellRanger v4.0.0 (10X Genomics) in the "count" pipeline using reference FASTA data form mice (mm10). As a result, 36,030 cells and 32,285 features passed through the quality control filter of CellRanger. Further analysis was conducted with the Seurat (v 3.2.3) R package (version 4.0.2). The cell counts generated in CellRanger were cut off based on the parameters - nUMI (nCount_RNA) > 500, nGene (nFeature_RNA) < 6000 and > 200, log10GenesPerUMI > 0.80, and mitoRatio < 0.20 - to remove possible cell doublets and potential apoptotic cells.

Filtering of the cells followed the integration protocol provided by Seurat to eliminate the batch effect of data generated from two different samples (Control vs. Dox)[46]. Using data normalized by condition through SCTransform() as input, 3000 features to be used for integration were selected using SelectIntegrationFeatures(). The functions FindIntegrationAnchors() and IntegrateData() with the method "SCT" as the normalization method were used to find cells with pairwise correlations between datasets (anchors) and integrate the two datasets by 'anchoring' the cells of each dataset in the same space. Integrated data were arranged through RunUMAP() after principal component analysis (PCA) dimensions were calculated with RunPCA(), and cell clustering was conducted with the FindNeighbors() and FindClusters() functions (clustering resolution = 0.1)

The major cell type of each cluster was identified by PanglaoDB (Supplementary Data 1)[47].

**RPE reanalysis**. When the threshold of *Rpe65*—the marker gene for RPE cells—was set to zero, all cells (36,030 cells) including RPE cells and putative contaminant cells showed *Rpe65* expression according to the Loupe browser. Among the 12 clusters of these cells in the UMAP plot, clusters 3 and 5 showed significantly

higher levels of Rpe65 than the others, and the other clusters showed expression of specific markers of cell types other than RPE cells (Supplementary Data 1).

Therefore, we speculated that the cells expressing low levels of *Rpe65* might not have been RPE cells. To obtain the RPE population, we separated the clusters with particularly high *Rpe65* gene expression (log2 FC > 2) from original cell dataset (36,030 cells) using the 'export' function of the Loupe browser and 'reanalyze' in CellRanger. The processing, filtering, integration, and clustering of the separated cells and features were performed in the same way as described for the whole dataset above.

For FindClusters(), clustering resolution of 0.05 was applied with default parameters. The cluster markers were identified by dividing the integrated data by condition and then classifying the cluster marker using FindAllMarkers() for each condition (with Wilcoxon's rank sum test).

The cell cycle (G1, G2M, S) phase of each cell was assigned through CellCycleScoring() using the phase information of the cycle.rda file provided by Seurat's official Vignette according to the standard protocol. This file contains a list of previously published canonical marker genes, enabling the calculation of cell cycle scores based on the expression values of genes. To check whether the bias of each parameter did not affect the formation of cluster 5, the distributions of nUMI, nGene, mitoRatio, and cell cycle score in each cluster and condition were visualized with UMAP plots (Supplementary Fig. 1).

The DEGs indicated condition-specific and cluster-specific expression, and all related parameters were calculated using FindMarkers() with Wilcoxon's rank sum test. An independent marker gene list was identified by separating the integrated data according to cluster and condition.

All of the UMAP plot and the cluster marker heatmap presented in the text were all visualized using the basic functions of the Seurat package.

**Functional enrichment of marker genes using GO enrichment analysis**. GO Biological Processes (http://geneontology.org/) were used to identify the function of each cluster marker genes (Figs. 2c and 3c) and DEGs (Fig. 5). The GO enrichment analysis of cluster markers was performed for the top 100 (logFC) marker genes for cluster (1–5) under each condition (Con vs. Dox).

**Canonical pathway analysis using IPA**. The list of marker genes of each RPE subcluster of control RPE and Dox-RPE cells were uploaded into the IPA software (QIAGEN, Germantown, MD, USA). The 'Core Analysis' function included in the software was used to interpret the canonical pathways of each RPE subcluster. Each gene identifier was mapped to its corresponding gene object in the Ingenuity Pathway Knowledge Base (IPKB).

**RNA fluorescence in situ hybridization in mouse RPE tissue**. To confirm the heterogeneity of RPE cells from the scRNAseq analyses, RNA fluorescence in situ hybridization was performed to visualize the heterogeneous distribution of RPE subclusters. For RNA detection in mouse tissues, we enucleated eye globes from euthanized mice, trimmed the muscle and fat surrounding the globes to create a smooth outer surface, and isolated the eyecup lined by the neural retina. Then, the neural retina was removed from the eyecup, and the remaining eyecup containing RPE/choroid/sclera tissues (RPE flatmount) was fixed in 4% paraformaldehyde/ PBS solution at 4 °C overnight. After fixation, the eyecups were subjected to in situ hybridization using an RNAscope multiplex fluorescent reagent kit (ACD, 320850) according to the manufacturer's protocol with slight modification. Briefly, the fixed eyecup was permeabilized with 0.1% TritonX-100/PBS for 15 min at room temperature (RT) and then treated with protease IV solution for 30 min at RT. Then, the eyecups were incubated at 40 °C in a HybEZ hybridization oven with target probes in hybridization buffer for 2 h. We used probes against mouse Rdh5 (ACD, 1152751-C1), Serping1 (ACD, 535071), Malat1 (ACD, 313391-C3), Cdkn1a (ACD, 408551-C2), A2m (ACD, 853411), and Bax (ACD, 463501) for the representative markers in clusters 1, 2, 3, 4, 4, 4, respectively, each of which showed distinct fold changes in the corresponding RPE subcluster in the scRNA seq data. Due to the limited wavelength of commercially available probes, Rdh5 and Serping1 could not be validated in the same RPE flatmount. The probes were amplified by each solution. The samples were incubated with Amp1 for 30 min at 40 °C and then washed twice in wash buffer for 2 min each. Amp2 was incubated on the samples for 15 min at 40 °C, followed by two washes in wash buffer. Then, the samples were incubated in Amp3 for 30 min at 40 °C and washed twice using wash buffer for 2 min each, followed by incubation of Amp4-Alt A at 40 °C for 30 min. Then, the nuclei were stained with DAPI for 15 min at RT. The eyecups were flat mounted on the slide glass with ProLong Gold (Invitrogen, P36930) mounting solution. The images were captured using a confocal laser scanning microscope (Carl Zeiss, LSM 900). Each acquired image from the RNAscope assay showed the spatial RNA expression of a specific maker gene.

Because three marker genes from three different subclusters were hybridized to corresponding probes in a single RPE flatmount, three images were generated from a single flatmount. Each 20-fold magnified image was exported to FIJI (US National Institutes of Health, Bethesda, Rockville, MD, USA), and the resolution was set to 1024 by 1024 (Fig. 6b). In each image, 'color threshold' was arbitrarily set and was applied to reduce the background noise signal and to clearly represent the heterogeneous distribution, resulting in a binarized image. Then, the whole area of this image was divided into 64 square areas. Each squared area was coded to number '1' or '0' according to the presence or absence, respectively, of any positive pixel inside the square area. To examine the spatial heterogeneity between two markers from the same RPE flatmount, the pattern of 64 binary sequences were statistically compared between two markers using McNemar's test. McNemar's test was used to compare proportions of the signal-positive and signal-negative square boxes between two markers in the same RPE flatmount, and $p < 0.05$ was interpreted as their expressing differently regarding spatial distribution, verifying the existence of subclusters from the analyses of scRNA-seq.

**Statistics and reproducibility**. Statistical analyses of the scRNA-seq data were carried out using the CellRanger and Seurat packages in R. In each cluster of RPE cell populations, the Wilcoxon rank sum test was used to perform gene expression comparisons between control and Dox-RPE cell clusters.

**Reporting summary**. Further information on research design is available in the Nature Research Reporting Summary linked to this article.

## Data availability

The scRNA-seq datasets utilized here have been deposited into the Gene Expression Omnibus (GEO) database (accession number GSE183572). All other data are available from the corresponding authors upon reasonable request.

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

## Acknowledgements

This study was supported by the National Research Foundation of Korea and funded by the Ministry of Science and ICT (NRF-2020M3A9D8038188, NRF-2019M3A9H1030948, and NRF-2020R1A2C2101941).

## Author contributions

H.C. and N.K. conceived, designed, and supervised the study and manuscript preparation. H.L. and H.L. (Hyungwoo Lee and Ho-Yeon Lee) contributed to the design, data analysis for most of the experiments and manuscript writing. J.C. contributed to the in vivo experiments. C.P., C.K., J.R, J.J. contributed to the data analysis and manuscript revision. All authors discussed the results and commented on the manuscript.

## Competing interests

The authors declare no competing interests.
