## [Peer Review File · Communications Biology]

Reviewers' comments:

Reviewer #1 (Remarks to the Author):

The author reported transcriptomic studies on Dox induced senescence RPEs in mice. By performing single cell RNA-seq on wild type and Dox-induced mice RPE, multiple cell type clusters including RPEs and many cell types in choroid are identified. DEG analysis between the wild type and Dox-induced RPEs are performed. DEGs that associated with apoptosis and senescence are observed. Overall this study provides a rich dataset with some interesting findings. The manuscript is well written and easy to follow. My specific comments are the following:

1. Five RPE subclusters are identified as in Figure 2. Among them, cluster 5 has very small number of cells and has high expression in muscle related genes. I am wondering how potential doublets are identified and removed? Is it possible the cluster 5 is primarily composed of doublet between RPE and muscle cell, for example. In addition, I am wondering if the authors have check if the PRE subclusters are potentially due to over cluster. UMI, mito ratio, cell cycle and other factors should be plotted on the UMAP plot to check if these clusters are driven by these confounding factors.
2. The title of figure two suggest the RPE clusters has "spatial and functional transcriptomic signatures". I am not sure where the spatial information comes from.
3. Since RPE cell in mature mice do not proliferate, I am wondering why a large portion of cell is in the S and G2M phase?
4. DEG analysis should be performed on the corresponding clusters. Therefore, cluster alignment or co-embedding should be performed to identify corresponding REP subtypes.
5. Since cell stress during tissue dissociation can lead to gene transcription changes, it is important to validate some of the key findings from scRNA-seq by orthogonal methods, such as RNAscope. In addition, the in situ RNA hybridization results will also provide important insights on spatial information of different RPE cell subtype or state information and the distribution of senescent RPEs in the RPE layer.

Reviewer #2 (Remarks to the Author):

In the previous single-cell transcriptome sequencing studies, the number of RPE cells obtained was small, as low as hundreds, and it was difficult to have a comprehensive understanding of RPE cells, especially in revealing their heterogeneity. This study collected 3355 mouse RPE cells, the analyzed RPE population is larger than previous studies, and analyzed the differences between RPE cell subgroups, which has a certain degree of innovation and research significance. This study used low-dose doxorubicin (Dox) to induce RPE senescence, and identified a subset of RPE cells that are susceptible to transcriptional changes, which can provide a reference for research on RPE cell senescence.

1. This study did not use sample data from AMD disease. Is it far-fetched to link Dox-induced RPE cell senescence with AMD, because Dox can induce relatively rapid changes in aging, but AMD is a slowly progressing disease. Affected and regulated by a variety of pathological factors, whether the transcriptional changes of the two cells are comparable remains to be studied.
2. There is no further experimental verification of the existence of RPE subgroups, especially cluster 4, a subgroup considered by the author to be most susceptible to aging and undergo transcriptional changes.
3. The visual effect is general and the form is single. For example, the heat maps of gene expression in cells use seurat's own functions without other modifications. I have seen exactly the same maps in other articles.
4. In terms of pathway enrichment, the author uses GO analysis, KEGG analysis or other more advanced analysis. It is recommended to find the genes that have a key role in each pathway to enhance persuasiveness, indicating that a certain subgroup is in a certain The function has a strong effect.

Reviewers' comments:
Reviewer #1

The author reported transcriptomic studies on Dox induced senescence RPEs in mice. By performing single cell RNA-seq on wild type and Dox-induced mice RPE, multiple cell type clusters including RPEs and many cell types in choroid are identified. DEG analysis between the wild type and Dox-induced RPEs are performed. DEGs that associated with apoptosis and senescence are observed. Overall this study provides a rich dataset with some interesting findings. The manuscript is well written and easy to follow. My specific comments are the following:

1. Five RPE subclusters are identified as in Figure 2. Among them, cluster 5 has very small number of cells and has high expression in muscle related genes. I am wondering how potential doublets are identified and removed? Is it possible the cluster 5 is primarily composed of doublet between RPE and muscle cell, for example. In addition, I am wondering if the authors have check if the PRE subclusters are potentially due to over cluster. UMI, mito ratio, cell cycle and other factors should be plotted on the UMAP plot to check if these clusters are driven by these confounding factors.

: To determine whether the presence of potential doublets in 3355 cells classified according to existing RPE65 expression affected the formation of the 5th cluster, we removed possible cell doublets by filtering out cells with more than 6000 expressed genes. Actually, we filtered out extraordinary cell data by applying the following criteria: nUMI (nCount_RNA) > 500, nGene (nFeature_RNA) < 6000 & >200, log10GenesPerUMI >0.80, and mitoRatio <0.20.

The elbow plot and the UMAP feature plot of each parameter including of the nUMI, nGene, mitoRatio, and cell cycle score values were generated to verify that the data were classified into an appropriate number of clusters and to confirm the existence of factors other than gene expression that could affect the formation of clusters (see Figure below).

As a result, the total number of clusters does not deviate significantly from the range of the optimal number of clusters shown in the elbow plot. Furthermore, parameters such as nFeature_RNA, nCount_RNA, and the cell scoring score do not show any bias toward a specific cluster; thus, we determined that the clustering result for the corresponding cell data was appropriate. We added this information to Supplementary Fig. 1.

As a result of performing clustering with filtered data through the same process used before, cluster 5, consisting of a small number of cells, was identified, as in the previous analysis results. Furthermore, GO analysis showed enrichment of muscle-related GO terms in the functional profile in this cluster, as shown in the existing unfiltered data.

Because the cell number was slightly decreased by doublet removal (N=3,355 → 3,228), we completely revised the RPE associated data (Table 1, Fig. 2, 3, 4, 5 and Supplementary Table 1, 2 and Supplementary data 2, 3, 4)

The two marker genes of cluster 5, Acta2 and Myl9 (Supplementary Table 1), have also been reported in cells other than muscle cells, such as a subpopulation of human pluripotent stem cells. The expression of these genes might be associated with active modulation of the cytoskeleton in various

cells undergoing cellular processes, such as epithelial-to-mesenchymal transition (EMT). The presence of the smooth muscle-associated genes—*Myl9* and *Mylk*—in cluster 5 might also be associated with the contractile potential of the RPE because during the induction of EMT of the RPE, increased contractility with upregulated expression of smooth muscle-related genes was observed. We added this discussion to page 17, line 712-718.

2. The title of figure two suggest the RPE clusters has “spatial and functional transcriptomic signatures”. I am not sure where the spatial information comes from.

: As you noted, the term ‘spatial’ is inappropriate based on these data. Thus, we deleted this term in Fig. 2.

3. Since RPE cell in mature mice do not proliferate, I am wondering why a large portion of cell is in the S and G2M phase?

: The cell cycle distribution shown in the figure below (Fig. 3c in the first submitted manuscript) was scored using the `CellCycleScoring()` function in the Seurat package. The cells were scored and assigned to three cell cycle phases, G1, S, and G2M, based on the marker genes of S and G2M phases.

<Fig. 3c in the first submitted manuscript>

Furthermore, the PCA analysis based on the states of G1, G2M and S phases showed overlapping distribution (Figure below. Left: UMAP plot of G1, G2M, S, respectively. Right: overlap of 3 UMAP plots).

As bias does not appear in a specific cell cycle distribution, the cell cycle phase does not seem to affect clustering.

Meanwhile, the CellCycleScoring() function does not support scoring for cell cycle phases other than the three specified above; thus, it cannot be considered to represent the current exact cell cycle distribution. To avoid confusion, we removed the original Fig. 3c (par plot shown above) from the main manuscript and added the UMAP plots, including cell cycle data, to Supplementary Fig. 1.

4. DEG analysis should be performed on the corresponding clusters. Therefore, cluster alignment or co-embedding should be performed to identify corresponding RPE subtypes.

: To confirm the expression pattern of DEGs for each cluster, the top 10 up-regulated and top 10 down-regulated DEGs after doxorubicin treatment were selected. Because not all of the top 10 up-regulated genes were statistically significantly significant ($p < 0.05$), fewer than 10 up-regulated DEGs were included. We added this result to Supplementary Fig. 3.

In addition, the comprehensive expression of these DEGs for each cluster was calculated using the AddModuleScore() function in Seurat to indicate the difference in expression by condition. We added this result to Supplementary Fig. 4.

Since some DEGs (Con vs. Dox in each cluster) overlapped in several clusters, functional analysis of each cluster was conducted using the cluster marker genes identified from the separated Con and Dox data.

5. Since cell stress during tissue dissociation can lead to gene transcription changes, it is important to validate some of the key findings from scRNA-seq by orthogonal methods, such as RNAscope. In addition, the in situ RNA hybridization results will also provide important insights on spatial information of different RPE cell subtype or state information and the distribution of senescent RPEs in the RPE layer.

: Thank you for this important comment. As you recommended, we performed in situ RNA hybridization to validate the scRNA-seq results. RNA fluorescence in situ hybridization was performed with the RNAscope assay on the flattened RPE (RPE flatmount) from control and Dox-treated mice. *Rdh5*, *Serping1*, and *Malat1* were the selected markers for RPE clusters 1, 2, and 3. For cluster 4, Dox-RPE-specific markers of apoptosis (*Bax1*) and senescence (*Cdkn1a*, *A2m*) were selected (Supplementary Table 1). Because of the insufficient number of RPEs in cluster 5 in both control RPE and Dox-RPE, we included only the markers of clusters 1 to 4 (Method section [page 21, line 919-944]).

For statistical verification of the heterogeneous distribution of each marker, each fluorescence image was divided into 64 binarized square tiles, and the presence or absence of signals in each tile was assessed. With this information, the McNemar test was performed to compare the similarity or difference in the spatial distribution between two markers (Fig. 6 middle and Method section [page 21, line 945-957]).

Rdh5, *Malat1* and *Serping1* showed differences in their spatial distribution with respect to each other, and the pattern of this spatial distribution was similar in both PBS- or Dox-injected areas and noninjected areas in the two groups (Control RPE and Dox-RPE, see Fig. 6, Supplementary Fig. 5). Three markers of Cluster 4 (*Bax1*, *Cdkn1a*, *A2m*) showed similar spatial distributions in the Dox-injected area in Dox-RPE, as expected, while their distribution was not superimposed with that of other cluster markers (Supplementary Fig. 5). We added these results to page 15, line 587-606, and discussed at page 18, line 737-751.

Therefore, we confirmed that our findings of transcriptomic changes—at least those in selected marker genes—are not derived mainly from the sample preparation process, including cellular stress during tissue dissociation. We also demonstrated that the RPE layer is composed of a heterogeneous population of cells, consistent with our scRNA-seq results. Thank you again for your valuable suggestion.

Reviewer #2 (Remarks to the Author):

In the previous single-cell transcriptome sequencing studies, the number of RPE cells obtained was small, as low as hundreds, and it was difficult to have a comprehensive understanding of RPE cells, especially in revealing their heterogeneity. This study collected 3355 mouse RPE cells, the analyzed RPE population is larger than previous studies, and analyzed the differences between RPE cell subgroups, which has a certain degree of innovation and research significance. This study used low-dose doxorubicin (Dox) to induce RPE senescence, and identified a subset of RPE cells that are susceptible to transcriptional changes, which can provide a reference for research on RPE cell senescence.

1. This study did not use sample data from AMD disease. Is it far-fetched to link Dox-induced RPE cell senescence with AMD, because Dox can induce relatively rapid changes in aging, but AMD is a slowly progressing disease. Affected and regulated by a variety of pathological factors, whether the transcriptional changes of the two cells are comparable remains to be studied.

: Thank you for your comment. It is our hope to use a better model that more accurately reproduces AMD, which progresses slowly over a long period of time; however the available mouse model of AMD remains limited. Certainly, this is one of the major limitations in AMD research. However, some animal models, including those used by our group, are considered useful for preclinical research on AMD. As you noted, Dox induces faster changes than the real progression of senescence anticipated in AMD. We agree with you that the RNA expression profile related to Dox-induced cellular senescence might not be comparable to that in AMD patients. In the present study, we performed scRNA-seq analysis in the mouse model of RPE senescence induced by Dox. In our recent study, we confirmed that this in vivo mouse model established by subretinal injection of Dox has some features similar to those of AMD, such as drusen-like subretinal deposits, autofluorescence and increased Bruch's membrane thickness. Furthermore, general features of cellular senescence—including increased SA- β -gal expression, increased cell size, activation of the senescence-associated secretory phenotype (SASP), and elevated expression of p53, p21, and p16—are observed in the Dox-RPE. Further laboratory research, such as the validation of drug effects, might be feasible in this model as shown in our recent study.

However, we completely agree with your opinion that our model cannot reproduce all the pathological findings and processes of AMD. We hope to characterize the transcriptome at the single-cell level in other mouse models based on complement factor pathway dysregulation, oxidative damage, and chronic conditions, for example, in aged mice over 24 months old. Finding similarities and differences between the control and model scRNA-seq data and comparing these data with scRNA-seq data from AMD patients might illuminate the detailed properties of AMD animal models and benefit researchers in designing future experiments. We added these to the discussion of limitations (page 18, line 781 to page 19, line 789).

2. There is no further experimental verification of the existence of RPE subgroups, especially cluster 4, a subgroup considered by the author to be most susceptible to aging and undergo transcriptional changes.

: As you recommended, we further validated the existence of RPE subgroups by performing fluorescence in situ RNA hybridization on flattened mouse RPE (RPE flatmount) in control and Dox-treated mice. The *Rdh5*, *Serping1*, and *Malat1* genes were selected for RPE clusters 1, 2, and 3, and Dox-RPE-specific markers of apoptosis (*Bax1*) and senescence (*Cdkn1a* and *A2m*) were selected for cluster 4 (page 21, line 919-944). Because of the paucity of RPEs in cluster 5, only clusters 1 to 4 were examined. McNemar's test was performed to statistically verify differences in spatial distribution between pairs of markers (Fig. 6 middle and Method section [page 21, line 945-957]).

Rdh5, *Malat1* and *Serping1* showed a heterogeneous spatial distribution, with each marker mainly found at different locations (Fig. 6 and Supplementary Fig. 5). This pattern was similar in both the PBS- and Dox-injected areas and the noninjected area in the two groups (Control RPE and Dox-RPE). The markers of cluster 4 of Dox-RPE (*Bax1*, *Cdkn1a*, and *A2m*) showed similar spatial distributions relative

to each other in the Dox-injected area in Dox-RPE, as expected, while their distribution was not superimposed with that of other cluster markers (Supplementary Fig. 5). We added these results to page 15, line 587-606, and discussed at page 18, line 737-751.

Collectively, these experimental results could validate the presence of RPE subpopulations. Thank you for your valuable advice.

3. The visual effect is general and the form is single. For example, the heat maps of gene expression in cells use Seurat's own functions without other modifications. I have seen exactly the same maps in other articles.

: Thank you for your comment. We agree with you that the graphical presentation should be modified. However, after discussion with the coauthors, we concluded that Seurat's Doheatmap() function should be used because it is a specialized method to show the marker gene expression in each cluster at the entire cluster level in our scRNA-seq data, as suggested in its own tutorial.

It would be better to change the style of the heatmaps and other figures to our own style, but we do not have the complete expertise to develop our own format. Thus, we concluded that no benefit could be gained by changing the format of either the existing heatmap or the existing violin plot.

Therefore, we simply changed the color palette of the heatmap to red and white and changed the shape of the violin plot for visual improvement. We apologize for this condition; we made our best effort to vary the visual effect. Thank you.

4. In terms of pathway enrichment, the author uses GO analysis, KEGG analysis or other more advanced analysis. It is recommended to find the genes that have a key role in each pathway to enhance persuasiveness, indicating that a certain subgroup is in a certain The function has a strong effect.

: Thank you for your insightful comment. In addition to GO analysis, pathway analysis was performed with QIAGEN's Ingenuity Pathway Analysis (IPA) approach for each RPE subcluster in control and Dox-RPE cells. Our analysis revealed a general overlap of canonical pathways between each cluster of control- and Dox-RPE cells. The 5 subclusters showed distinct pathways, and the apoptosis pathway was identified only in cluster 4 of Dox-RPE cells, confirming that the transcriptome of cluster 4 of Dox-RPE cells represents the senescence-specific gene expression. We added this information to the Results section (page 9, line 387-392), Methods section (page 21, line 913-917) and Bar plots of the canonical pathways are presented in Supplementary Fig. 2.

REVIEWERS' COMMENTS:

Reviewer #1 (Remarks to the Author):

The authors have adequately addressed critics raised by the reviewers. The manuscript is ready to be accepted.

Reviewer #2 (Remarks to the Author):

In this study, the transcriptional profiles of 3355 RPE cells were obtained by single-cell RNA sequencing. The number of RPE cells analyzed was more than that in previous studies, which is of great significance for a comprehensive and in-depth understanding of RPE cells. The authors used low dose doxorubicin (Dox) to induce RPE senescence, and analyzed the differences between RPE cell subsets using multiple data analysis methods. Furthermore, the presence of RPE cell subsets were validated using fluorescence in situ RNA hybridization. In particular, a subset of RPE cells that are susceptible to senescence were identified, which has certain innovation and research value, and can provide a reference for senescence research of RPE cells.